# Learning brain regions via large-scale online structured sparse dictionary-learning

**Elvis Dohmatob, Arthur Mensch, Gael Varoquaux, Bertrand Thirion**
`firstname.lastname@inria.fr`
Parietal Team, INRIA / CEA, Neurospin, Université Paris-Saclay, France

## Abstract

We propose a multivariate online dictionary-learning method for obtaining decompositions of brain images with structured and sparse components (aka atoms). Sparsity is to be understood in the usual sense: the dictionary atoms are constrained to contain mostly zeros. This is imposed via an $\ell_1$-norm constraint. By "structured", we mean that the atoms are piece-wise smooth and compact, thus making up blobs, as opposed to scattered patterns of activation. We propose to use a Sobolev (Laplacian) penalty to impose this type of structure. Combining the two penalties, we obtain decompositions that properly delineate brain structures from functional images. This non-trivially extends the online dictionary-learning work of Mairal et al. (2010), at the price of only a factor of 2 or 3 on the overall running time. Just like the Mairal et al. (2010) reference method, the online nature of our proposed algorithm allows it to scale to arbitrarily sized datasets. Preliminary xperiments on brain data show that our proposed method extracts structured and denoised dictionaries that are more intepretable and better capture inter-subject variability in small medium, and large-scale regimes alike, compared to state-of-the-art models.

## 1 Introduction

In neuro-imaging, inter-subject variability is often handled as a statistical residual and discarded. Yet there is evidence that it displays structure and contains important information. Univariate models are ineffective both computationally and statistically due to the large number of voxels compared to the number of subjects. Likewise, statistical analysis of weak effects on medical images often relies on defining regions of interests (ROIs). For instance, pharmacology with Positron Emission Tomography (PET) often studies metabolic processes in specific organ sub-parts that are defined from anatomy. Population-level tests of tissue properties, such as diffusion, or simply their density, are performed on ROIs adapted to the spatial impact of the pathology of interest. Also, in functional brain imaging, e.g function magnetic resonance imaging (fMRI), ROIs must be adapted to the cognitive process under study, and are often defined by the very activation elicited by a closely related process [18]. ROIs can boost statistical power by reducing multiple comparisons that plague image-based statistical testing. If they are defined to match spatially the differences to detect, they can also improve the signal-to-noise ratio by averaging related signals. However, the crux of the problem is how to define these ROIs in a principled way. Indeed, standard approaches to region definition imply a segmentation step. Segmenting structures in individual statistical maps, as in fMRI, typically yields meaningful units, but is limited by the noise inherent to these maps. Relying on a different imaging modality hits cross-modality correspondence problems.

**Sketch of our contributions.** In this manuscript, we propose to use the *variability* of the statistical maps across the population to define regions. This idea is reminiscent of clustering approaches, that have been employed to define spatial units for quantitative analysis of information as diverse as brain fiber tracking, brain activity, brain structure, or even imaging-genetics. See [21, 14] and references therein. The key idea is to group together features –voxels of an image, vertices on a mesh, fiber tracts–

based on the quantity of interest, to create regions –or fiber bundles– for statistical analysis. However, unlike clustering that models each observation as an instance of a cluster, we use a model closer to the signal, where each observation is a linear mixture of several signals. The model is closer to mode finding, as in a principal component analysis (PCA), or an independent component analysis (ICA), often used in brain imaging to extract functional units [5]. Yet, an important constraint is that the modes should be sparse and spatially-localized. For this purpose, the problem can be reformulated as a linear decomposition problem like ICA/PCA, with appropriate spatial and sparse penalties [25, 1].

We propose a multivariate online dictionary-learning method for obtaining decompositions with structured and sparse components (aka atoms). Sparsity is to be understood in the usual sense: the atoms contain mostly zeros. This is imposed via an $\ell_1$ penalty on the atoms. By "structured", we mean that the atoms are piece-wise smooth and compact, thus making up blobs, as opposed to scattered patterns of activation. We impose this type of structure via a Laplacian penalty on the dictionary atoms. Combining the two penalties, we therefore obtain decompositions that are closer to known functional organization of the brain. This non-trivially extends the online dictionary-learning work [16], with only a factor of 2 or 3 on the running time. By means of experiments on a large public dataset, we show the improvements brought by the spatial regularization with respect to traditional $\ell_1$-regularized dictionary learning. We also provide a concise study of the impact of hyper-parameter selection on this problem and describe the optimality regime, based on relevant criteria (reproducibility, captured variability, explanatory power in prediction problems).

## 2 Smooth Sparse Online Dictionary-Learning (Smooth-SODL)

Consider a stack $\mathbf{X} \in \mathbb{R}^{n \times p}$ of $n$ subject-level brain images $\mathbf{X}_1, \mathbf{X}_2, \ldots, \mathbf{X}_n$ each of shape $n_1 \times n_2 \times n_3$, seen as $p$-dimensional row vectors –with $p = n_1 \times n_2 \times n_3$, the number of voxels. These could be images of fMRI activity patterns like statistical parametric maps of brain activation, raw pre-registered (into a common coordinate space) fMRI time-series, PET images, etc. We would like to decompose these images as a mixture of $k \leq \min(n, p)$ component maps (aka latent factors or dictionary atoms) $\mathbf{V}^1, \ldots, \mathbf{V}^k \in \mathbb{R}^{p \times 1}$ and modulation coefficients $\mathbf{U}_1, \ldots, \mathbf{U}_n \in \mathbb{R}^{k \times 1}$ called *codes* (one $k$-dimensional code per sample point), i.e

$$\mathbf{X}_i \approx \mathbf{V}\mathbf{U}_i, \text{ for } i = 1, 2, \ldots, n \tag{1}$$

where $\mathbf{V} := [\mathbf{V}^1 | \ldots | \mathbf{V}^k] \in \mathbb{R}^{p \times k}$, an unknown dictionary to be estimated. Typically, $p \sim 10^5 - 10^6$ (in full-brain high-resolution fMRI) and $n \sim 10^2 - 10^5$ (for example, in considering all the 500 subjects and all the about functional tasks of the Human Connectome Project dataset [20]). Our work handles the extreme case where both $n$ and $p$ are large (massive-data setting). It is reasonable then to only consider under-complete dictionaries: $k \leq \min(n, p)$. Typically, we use $k \sim 50$ or $100$ components. It should be noted that online optimization is not only crucial in the case where $n/p$ is big; it is relevant whenever $n$ is large, leading to prohibitive memory issues irrespective of how big or small $p$ is.

As explained in section 1, we want the component maps (aka dictionary atoms) $\mathbf{V}^j$ to be sparse and spatially smooth. A principled way to achieve such a goal is to impose a boundedness constraint on $\ell_1$-like norms of these maps to achieve sparsity and simultaneously impose smoothness by penalizing their Laplacian. Thus, we propose the following penalized dictionary-learning model

$$\min_{\mathbf{V} \in \mathbb{R}^{p \times k}} \left( \lim_{n \to \infty} \frac{1}{n} \sum_{i=1}^{n} \min_{\mathbf{U}_i \in \mathbb{R}^k} \frac{1}{2} \|\mathbf{X}_i - \mathbf{V}\mathbf{U}_i\|_2^2 + \frac{1}{2}\alpha\|\mathbf{U}_i\|_2^2 \right) + \gamma \sum_{j=1}^{k} \Omega_{\text{Lap}}(\mathbf{V}^j). \tag{2}$$

subject to $\mathbf{V}^1, \ldots, \mathbf{V}^k \in \mathcal{C}$

The ingredients in the model can be broken down as follows:

- Each of the terms $\max_{\mathbf{U}_i \in \mathbb{R}^k} \frac{1}{2}\|\mathbf{X}_i - \mathbf{V}\mathbf{U}_i\|_2^2$ measures how well the current dictionary $\mathbf{V}$ explains data $\mathbf{X}_i$ from subject $i$. The Ridge penalty term $\phi(\mathbf{U}_i) \equiv \frac{1}{2}\alpha\|\mathbf{U}_i\|_2^2$ on the codes amounts to assuming that the energy of the decomposition is spread across the different samples. In the context of a specific neuro-imaging problem, if there are good grounds to assume that each sample / subject should be sparsely encoded across only a few atoms of the dictionary, then we can use the $\ell_1$ penalty $\phi(\mathbf{U}_i) := \alpha\|\mathbf{U}_i\|_1$ as in [16]. We note that in

contrast to the $\ell_1$ penalty, the Ridge leads to stable codes. The parameter $\alpha > 0$ controls the amount of penalization on the codes.

- The constraint set $\mathcal{C}$ is a sparsity-inducing compact simple (mainly in the sense that the Euclidean projection onto $\mathcal{C}$ should be easy to comput) convex subset of $\mathbb{R}^p$ like an $\ell_1$-ball $\mathbb{B}_{p,\ell_1}(\tau)$ or a simplex $\mathcal{S}_p(\tau)$, defined respectively as

$$\mathbb{B}_{p,\ell_1}(\tau) := \left\{\mathbf{v} \in \mathbb{R}^p \text{ s.t } |\mathbf{v}_1| + \ldots + |\mathbf{v}_p| \le \tau\right\}, \text{ and } \mathcal{S}_p(\tau) := \mathbb{B}_{p,\ell_1}(\tau) \cap \mathbb{R}_+^p. \quad (3)$$

Other choices (e.g ElasticNet ball) are of course possible. The radius parameter $\tau > 0$ controls the amount of sparsity: smaller values lead to sparser atoms.

- Finally, $\Omega_{\text{Lap}}$ is the 3D Laplacian regularization functional defined by

$$\Omega_{\text{Lap}}(\mathbf{v}) := \frac{1}{2}\sum_{k=1}^{p}(\nabla_x\mathbf{v})_k^2 + (\nabla_y\mathbf{v})_k^2 + (\nabla_z\mathbf{v})_k^2 = \frac{1}{2}\mathbf{v}^T\Delta\mathbf{v} \ge 0, \ \forall \mathbf{v} \in \mathbb{R}^p, \quad (4)$$

$\nabla_x$ being the discrete spatial gradient operator along the $x$-axis (a $p$-by-$p$ matrix), $\nabla_y$ along the $y$-axis, etc., and $\Delta := \nabla^T\nabla$ is the $p$-by-$p$ matrix representing the discrete Laplacian operator. This penalty is meant to impose blobs. The regularization parameter $\gamma \ge 0$ controls how much regularization we impose on the atoms, compared to the reconstruction error.

The above formulation, which we dub *Smooth Sparse Online Dictionary-Learning* (Smooth-SODL) is inspired by, and generalizes the standard online dictionary-learning framework of [16] –henceforth referred to as *Sparse Online Dictionary-Learning* (SODL)– with corresponds to the special case $\gamma = 0$.

## 3 Estimating the model

### 3.1 Algorithms

The objective function in problem (2) is separately convex and block-separable w.r.t each of $\mathbf{U}$ and $\mathbf{V}$ but is not jointly convex in $(\mathbf{U}, \mathbf{V})$. Also, it is continuously differentiable on the constraint set, which is compact and convex. Thus by classical results (e.g Bertsekas [6]), the problem can be solved via Block-Coordinate Descent (BCD) [16]. Reasoning along the lines of [15], we derive that the BCD iterates are as given in Alg. 1 in which, for each incoming sample point $\mathbf{X}_t$, the loading vector $\mathbf{U}_t$ is computing by solving a ridge regression problem (5) with the current dictionary $\mathbf{V}_t$ held fixed, and the dictionary atoms are then updated sequentially via Alg. 2. A crucial advantage of using a BCD scheme is that it is parameter free: there is not step size to tune. The resulting algorithm Alg. 1, is adapted from [16]. It relies on Alg. 2 for performing the structured dictionary updates, the details of which are discussed below.

---

**Algorithm 1** Online algorithm for the dictionary-learning problem (2)

---

**Require:** Regularization parameters $\alpha, \gamma > 0$; initial dictionary $\mathbf{V} \in \mathbb{R}^{p \times k}$, number of passes / iterations $T$ on the data.
1: $\mathbf{A}_0 \leftarrow 0 \in \mathbb{R}^{k \times k}$, $\mathbf{B}_0 \leftarrow 0 \in \mathbb{R}^{p \times k}$ (historical "sufficient statistics")
2: **for** $t = 1$ to $T$ **do**
3:     Empirically draw a sample point $\mathbf{X}_t$ at random.
4:     Code update: Ridge-regression (via SVD of current dictionary $\mathbf{V}$)

$$\mathbf{U}_t \leftarrow \text{argmin}_{\mathbf{u} \in \mathbb{R}^k} \frac{1}{2}\|\mathbf{X}_t - \mathbf{V}\mathbf{u}\|_2^2 + \frac{1}{2}\alpha\|\mathbf{u}\|_2^2. \quad (5)$$

5:     Rank-1 updates: $\mathbf{A}_t \leftarrow \mathbf{A}_{t-1} + \mathbf{U}_t\mathbf{U}_t^T$, $\mathbf{B}_t \leftarrow \mathbf{B}_{t-1} + \mathbf{X}_t\mathbf{U}_t^T$
6:     BCD dictionary update: Compute update for dictionary $\mathbf{V}$ using Alg. 2.
7: **end for**

---

**Update of the codes: Ridge-coding.** The Ridge sub-problem for updating the codes

$$\mathbf{U}_t = (\mathbf{V}^T\mathbf{V} + \alpha\mathbf{I})^{-1}\mathbf{V}^T\mathbf{X}_t \quad (6)$$

is computed via an SVD of the current dictionary $\mathbf{V}$. For $\alpha \approx 0$, $\mathbf{U}_t$ reduces to the orthogonal projection of $\mathbf{X}_t$ onto the image of the current dictionary $\mathbf{V}$. As in [16], we speed up the overall algorithm by sampling mini-batches of $\eta$ samples $\mathbf{X}_t, \ldots, \mathbf{X}_\eta$ and compute the corresponding codes $\mathbf{U}_1, \mathbf{U}_2, ..., \mathbf{U}_\eta$ at once. We typically use we use mini-batches of size $\eta = 20$.

**BCD dictionary update for the dictionary atoms.** Let us define time-varying matrices $\mathbf{A}_t := \sum_{i=1}^t \mathbf{U}_i \mathbf{U}_i^T \in \mathbb{R}^{k \times k}$ and $\mathbf{B}_t := \sum_{i=1}^t \mathbf{X}_i \mathbf{U}_i^T \in \mathbb{R}^{p \times k}$, where $t = 1, 2, \ldots$ denotes time. We fix the matrix of codes $\mathbf{U}$, and for each $j$, consider the update of the $j$th dictionary atom, with all the other atoms $\mathbf{V}^{k \neq j}$ kept fixed. The update for the atom $\mathbf{V}^j$ can then be written as

$$
\mathbf{V}^j = \mathrm{argmin}_{\mathbf{v} \in \mathcal{C}, \mathbf{V} = [\mathbf{V}^1 | \ldots | \mathbf{v} | \ldots | \mathbf{V}^k]} \left( \sum_{i=1}^t \frac{1}{2} \|\mathbf{X}_i - \mathbf{V}\mathbf{U}_i\|_2^2 \right) + \gamma t \Omega_{\mathrm{Lap}}(\mathbf{v})
$$

$$
= \mathrm{argmin}_{\mathbf{v} \in \mathcal{C}} F_{\gamma(\mathbf{A}_t[j,j]/t)^{-1}}(\mathbf{v}, \underbrace{\mathbf{V}^j + \mathbf{A}_t[j,j]^{-1}(\mathbf{B}_t^j - \mathbf{V}\mathbf{A}_t^j)}_{\text{refer to [16] for the details}}),
\tag{7}
$$

where $F_{\tilde{\gamma}}(\mathbf{v}, \mathbf{a}) \equiv \frac{1}{2}\|\mathbf{v} - \mathbf{a}\|_2^2 + \tilde{\gamma} \Omega_{\mathrm{Lap}}(\mathbf{v}) = \frac{1}{2}\|\mathbf{v} - \mathbf{a}\|_2^2 + \frac{1}{2}\tilde{\gamma}\mathbf{v}^T \Delta \mathbf{v}$.

---

**Algorithm 2** BCD dictionary update with Laplacian prior

---

**Require:** $\mathbf{V} = [\mathbf{V}^1 | \ldots | \mathbf{V}^k] \in \mathbb{R}^{p \times k}$ (input dictionary),
1: $\mathbf{A} = [\mathbf{A}^1 | \ldots | \mathbf{A}^k] \in \mathbb{R}^{k \times k}$, $\mathbf{B}_t = [\mathbf{B}_t^1 | \ldots | \mathbf{B}_t^k] \in \mathbb{R}^{p \times k}$ (history)
2: **while** stopping criteria not met, **do**
3:     **for** $j = 1$ to $r$ **do**
4:         Fix the code $\mathbf{U}$ and all atoms $k \neq j$ of the dictionary $\mathbf{V}$ and then update $\mathbf{V}^j$ as follows

$$
\mathbf{V}^j \leftarrow \mathrm{argmin}_{\mathbf{v} \in \mathcal{C}} F_{\gamma(\mathbf{A}_t[j,j]/t)^{-1}}(\mathbf{v}, \mathbf{V}^j + \mathbf{A}_t[j,j]^{-1}(\mathbf{B}_t^j - \mathbf{V}\mathbf{A}_j))
\tag{8}
$$

        (See below for details on the derivation and the resolution of this problem)
5:     **end for**
6: **end while**

---

Problem (7) is the compactly-constrained minimization of the 1-strongly-convex quadratic functions $F_{\tilde{\gamma}}(., \mathbf{a}) : \mathbb{R}^p \to \mathbb{R}$ defined above. This problem can further be identified with a denoising instance (i.e in which the design matrix / deconvolution operator is the identity operator) of the GraphNet model [11, 13]. Fast first-order methods like FISTA [4] with optimal rates $\mathcal{O}(L/\sqrt{\epsilon})$ are available[1] for solving such problems to arbitrary precision $\epsilon > 0$. One computes the Lipschitz constant to be $L_{F_{\tilde{\gamma}}(., \mathbf{a})} \equiv 1 + \tilde{\gamma} L_{\Omega_{\mathrm{Lap}}} = 1 + 4D\tilde{\gamma}$, where as before, $D$ is the number of spatial dimensions ($D = 3$ for volumic images). One should also mention that under certain circumstances, it is possible to perform the dictionary updates in the Fourier domain, via FFT. This alternative approach is detailed in the supplementary materials.

Finally, one notes that, since constraints in problem (2) are separable in the dictionary atoms $\mathbf{V}^j$, the BCD dictionary-update algorithm Alg. 2 is guaranteed to converge to a global optimum, at each iteration [6, 16].

**How difficult is the dictionary update for our proposed model ?** A favorable property of the vanilla dictionary-learning [16] is that the BCD dictionary updates amount to Euclidean projections onto the constraint set $\mathcal{C}$, which can be easily computed for a variety of choices (simplexes, closed convex balls, etc.). One may then ask: do we retain a comparable algorithmic simplicity even with the additional Laplacian terms $\Omega_{\mathrm{Lap}}(\mathbf{V}^j)$ ? **YES!**: empirically, we found that 1 or 2 iterations of FISTA [4] are sufficient to reach an accuracy of $10^{-6}$ in problem (7), which is sufficient to obtain a good decomposition in the overall algorithm.

However, choosing $\gamma$ "too large" will provably cause the dictionary updates to eventually take forever to run. Indeed, the Lipschitz constant in problem (7) is $L_t = 1 + 4D\gamma(\mathbf{A}_t[j,j]/t)^{-1}$, which will blow-up (leading to arbitrarily small step-sizes) unless $\gamma$ is chosen so that

$$
\gamma = \gamma_t = \mathcal{O}\left( \max_{1 \leq j \leq k} \mathbf{A}_t[j,j] \right) = \mathcal{O}\left( \max_{1 \leq j \leq k} \sum_{i=1}^t \|\mathbf{U}^j\|_2^2 / t \right) = \mathcal{O}(\|A_t\|_{\infty, \infty}/t).
\tag{9}
$$

Finally, the Euclidean projections onto the $\ell_1$ ball $\mathcal{C}$ can be computed exactly in linear-time $\mathcal{O}(p)$ (see for example [7, 9]). The dictionary atoms $j$ are repeatedly cycled and problem (7) solved. All in all, in practice we observe that a single iteration is sufficient for the dictionary update sub-routine in Alg. 2 to converge to a qualitatively good dictionary.

**Convergence of the overall algorithm.** The Convergence of our algorithm (to a local optimum) is guaranteed since all hypotheses of [16] are satisfied. For example, assumption (**A**) is satisfied because fMRI data are naturally compactly supported. Assumption (**C**) is satisfied since the ridge-regression problem (5) has a unique solution. More details are provided in the supplementary materials.

## 3.2 Practical considerations

**Hyper-parameter tuning.** Parameter-selection in dictionary-learning is known to be a difficult unsolved problem [16, 15], and our proposed model (2) is not an exception to this rule. We did an extensive study of the quality of estimated dictionary varies with the model hyper-parameters $(\alpha, \gamma, \tau)$. The data experimental setup is described in Section 5. The results are presented in Fig. 1. We make the following observations: Taking the sparsity parameter $\tau$ in (2) too large leads to dense atoms that perfectly explain the data but are not very intepretable. Taking it too small leads to overly sparse maps that barely explain the data. This normalized sparsity metric (small is better, *ceteris paribus*) is defined as the mean ratio $\|\mathbf{V}^j\|_1 / \|\mathbf{V}^j\|_2$ over the dictionary atoms.

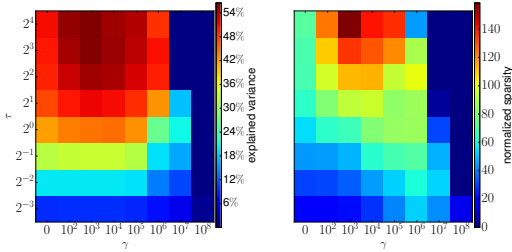

Figure 1: **Influence of model parameters.** In the experiments, $\alpha$ was chosen according to (10). **Left:** Percentage explained variance of the decomposition, measured on left-out data split. **Right:** Average normalized sparsity of the dictionary atoms.

Concerning the $\alpha$ parameter, inspired by [26], we have found the following time-varying data-adaptive choice for the $\alpha$ parameter to work very well in practice:

$$\alpha = \alpha_t \sim t^{-1/2}. \tag{10}$$

Likewise, care must be taken in selecting the Laplacian regularization parameter $\gamma$. Indeed taking it too small amounts to doing vanilla dictionary-learning model [16]. Taking it too large can lead to degenerate maps, as the spatial regularization then dominates the reconstruction error (data fidelity) term. We find that there is a safe range of the parameter pair $(\gamma, \tau)$ in which a good compromise between the sparsity of the dictionary (thus its intepretability) and its explanation power of the data can be reached. See Fig. 1. $K$-fold cross-validation with explained variance metric was retained as a good strategy for setting the Laplacian regularization $\gamma$ parameter and the sparsity parameter $\tau$.

**Initialization of the dictionary.** Problem (2) is non-convex jointly in $(\mathbf{U}, \mathbf{V})$, and so initialization might be a crucial issue. However, in our experiments, we have observed that even randomly initialized dictionaries eventually produce sensible results that do not jitter much across different runs of the same experiment.

## 4 Related works

While there exist algorithms for online sparse dictionary-learning that are very efficient in large-scale settings (for example [16], or more recently [17]) imposing spatial structure introduces couplings in the corresponding optimization problem [8]. So far, spatially-structured decompositions have been solved by very slow alternated optimization [25, 1]. Notably, structured priors such as TV-$\ell_1$ [3] minimization, were used by [1] to extract data-driven state-of-the-art atlases of brain function. However, alternated minimization is very slow, and large-scale medical imaging has shifted to online solvers for dictionary-learning like [16] and [17]. These do not readily integrate structured penalties. As a result, the use of structured decompositions has been limited so far, by the computational cost of the resulting algorithms. Our approach instead uses a Laplacian penalty to impose spatial structure at

a very minor cost and adapts the online-learning dictionary-learning framework [16], resulting in a fast and scalable structured decomposition. Second, the approach in [1] though very novel, is mostly heuristic. In contrast, our method enjoys the same convergence guarantees and comparable numerical complexity as the basic unstructured online dictionary-learning [16].

Finally, one should also mention [23] that introduced an online group-level functional brain mapping strategy for differentiating regions reflecting the variety of brain network configurations observed in a the population, by learning a sparse-representation of these in the spirit of [16].

## 5   Experiments

**Setup.**   Our experiments were done on task fMRI data from 500 subjects from the HCP –Human Connectome Project– dataset [20]. These task fMRI data were acquired in an attempt to assess major domains that are thought to sample the diversity of neural systems of interest in functional connectomics. We studied the activation maps related to a task that involves language (story understanding) and mathematics (mental computation). This particular task is expected to outline number, attentional and language networks, but the variability modes observed in the population cover even wider cognitive systems. For the experiments, mass-univariate General Linear Models (GLMs) [10] for $n = 500$ subjects were estimated for the *Math vs Story* contrast (language protocol), and the corresponding full-brain $Z$-score maps each containing $p = 2.6 \times 10^5$ voxels, were used as the input data $\mathbf{X} \in \mathbb{R}^{n \times p}$, and we sought a decomposition into a dictionary of $k = 40$ atoms (components). The input data $\mathbf{X}$ were shuffled and then split into two groups of the same size.

**Models compared and metrics.**   We compared our proposed Smooth-SODL model (2) against both the Canonical ICA –CanICA [22], a single-batch multi-subject PCA/ICA-based method, and the standard SODL (sparse online dictionary-learning) [16]. While the CanICA model accounts for subject-to-subject differences, one of its major limitations is that it does not model spatial variability across subjects. Thus we estimated the CanICA components on smoothed data: isotropic FWHM of 6mm, a necessary preprocessing step for such methods. In contrast, we did not perform pre-smoothing for the SODL of Smooth-SODL models. The different models were compared across a variety of qualitative and quantitative metrics: visual quality of the dictionaries obtained, explained variance, stability of the dictionary atoms, their reproducibility, performance of the dictionaries in predicting behavioral scores (IQ, picture vocabulary, reading proficiency, etc.) shipped with the HCP data [20]. For both SODL [16] and our proposed Smooth-SODL model, the constraint set for the dictionary atoms was taken to be a simplex $\mathcal{C} := \mathcal{S}_p(\tau)$ (see section 2 for definition). The results of these experiments are presented in Fig. 2 and Tab. 1.

## 6   Results

**Running time.**   On the computational side, the vanilla dictionary-learning SODL algorithm [16] with a batch size of $\eta = 20$ took about 110s ($\approx 1.7$ minutes) to run, whilst with the same batch size, our proposed Smooth-SODL model (2) implemented in Alg. 1 took 340s ($\approx 5.6$ minutes), which is slightly less than **3 times** slower than SODL. Finally, CanICA [22] for this experiment took 530s ($\approx 8.8$ minutes) to run, which is about **5 times** slower than the SODL model and **1.6 times** slower than our proposed Smooth-SODL (2) model. All experiments were run on a single CPU of laptop.

**Qualitative assessment of dictionaries.**   As can be seen in Fig. 2*(a)*, all methods recover dictionary atoms that represent known functional brain organization; notably the dictionaries all contain the well-known executive control and attention networks, at least in part. Vanilla dictionary-learning leverages the denoising properties of the $\ell_1$ sparsity constraint, but the voxel clusters are not very structured. For, example most blobs are surrounded with a thick ring of very small nonzero values. In contrast, our proposed regularization model leverages both sparse and structured dictionary atoms, that are more spatially structured and less noisy.

In contrast to both SODL and Smooth-SODL, CanICA [22] is an ICA-based method that enforces no notion of sparsity whatsoever. The result are therefore dense and noisy dictionary atoms that explain the data very well (Fig. 2*(b)* but which are completely uninterpretable. In a futile attempt to remedy the situation, in practice such PCA/ICA-based methods (including FSL's MELODIC tool [19]) are hard-thresholded in order to see information. For CanICA, the hard-thresholded version has been

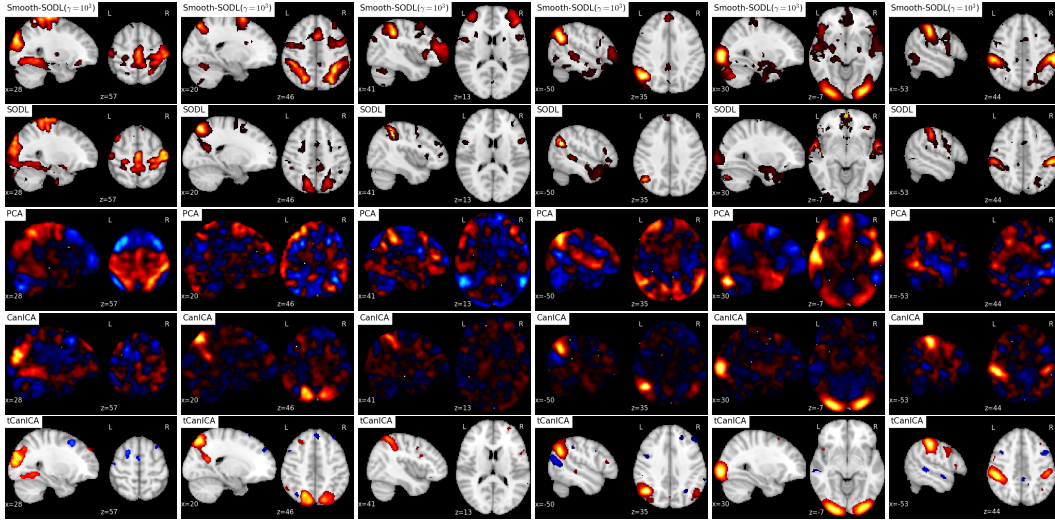

(a) **Qualitative comparison of the estimated dictionaries.** Each column represents an atom of the estimated dictionary, where atoms from the different models (the rows of the plots) have been matched via a Hungarian algorithm. Here, we only show a limited number of the most "intepretable" atoms. Notice how the major structures in each atom are reproducible across the different models. Maps corresponding to hard-thresholded CanICA [22] components have also been included, and have been called tCanICA. In contrast, the maps from the SODL [16] and our proposed Smooth-SODL (2) have not been thresholded.

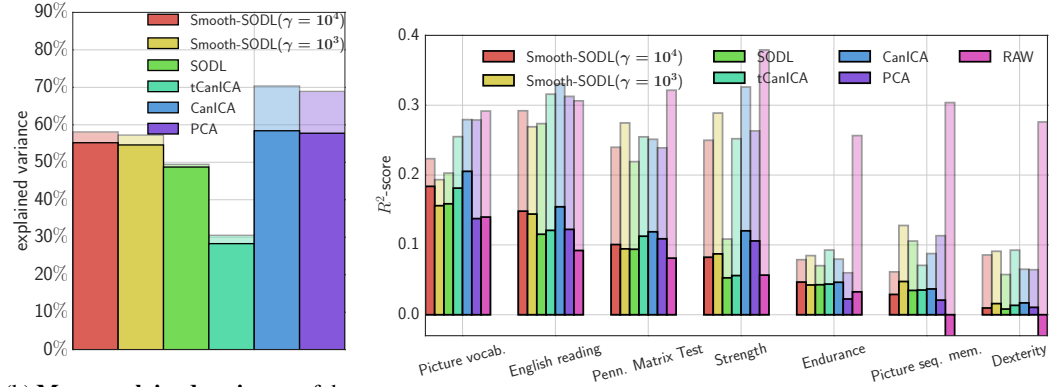

(b) **Mean explained variance** of the different models on both training data and test (left-out) data. **N.B.:** Bold bars represent performance on **test** set while faint bars in the background represent performance on **train** set.

(c) **Predicting behavioral variables** of the HCP [20] dataset using subject-level $Z$-maps. **N.B.:** Bold bars represent performance on **test** set while faint bars in the background represent performance on **train** set.

Figure 2: **Main results.** Benchmarking our proposed Smooth-SODL (2) model against competing state-of-the-art methods like SODL (sparse online dictionary-learning) [16] and CanICA [22].

named tCanICA in Fig. 2. That notwithstanding, notice how the major structures (parietal lobes, sulci, etc.) in each atom are reproducible across the different models.

**Stability-fidelity trade-offs.** PCA/ICA-based methods like CanICA [22] and MELODIC [19] are the optimal linear decomposition method to maximize explained variance on a dataset. On the training set, CanICA [22] out-performs all others algorithms with about 66% (resp. 50% for SODL [16] and 58% for Smooth-SODL) of explained variance on the training set, and 60% (resp. 49% for SODL and **55%** for Smooth-SODL) on left-out (test) data. See Fig. 2(b). However, as noted in the above paragraph, such methods lead to dictionaries that are hardly intepretable and thus the user must recourse to some kind of post-processing hard-thresholding step, which destroys the estimated model. More so, assessing the stability of the dictionaries, measured by mean correlation between corresponding atoms, across different splits of the data, CanICA [22] scores a meager 0.1, whilst the hard-thresholded version tCanICA obtains 0.2, compared to **0.4** for Smooth-SODL and 0.1 for SODL.

**Is spatial regularization really needed ?** As rightly pointed out by one of the reviewers, one does not need spatial regularization if data are abundant (like in the HCP). So we computed learning curves of mean explained variance (EV) on test data, as a function of the amount training data seen by both Smooth-SODL and SODL [16] (Table 1). In the beginning of the curve, our proposed spatially regularized Smooth-SODL model starts off with more than 31% explained variance (computed on 241 subjects), after having pooled only 17 subjects. In contrast, the vanilla SODL model [16] scores a meager 2% explained variance; this corresponds to a 14-fold gain of Smooth-SODL over SODL. As more and more data are pooled, both models explain more variance, the gap between Smooth-SODL and SODL reduces, and both models perform comparably asymptotically.

| Nb. subjects pooled | mean EV for vanilla SODL | Smooth-SODL (2) | gain factor |
|:---:|:---:|:---:|:---:|
| 17 | **2%** | **31%** | **13.8** |
| 92 | 37% | 50% | 1.35 |
| 167 | 47% | 54% | 1.15 |
| 241 | 49% | **55%** | 1.11 |

Table 1: **Learning-curve** for boost in explained variance of our proposed Smooth-SODL model over the reference SODL model. Note the reduction in the explained variance gain as more data are pooled.

Thus our proposed Smooth-SODL method extracts structured denoised dictionaries that better capture inter-subject variability in small, medium, and large-scale regimes alike.

**Prediction of behavioral variables.** If Smooth-SODL captures the patterns of inter-subject variability, then it should be possible to predict cognitive scores $\mathbf{y}$ like picture vocabulary, reading proficiency, math aptitude, etc. (the behavioral variables are explained in the HCP wiki [12]) by projecting new subjects' data into this learned low-dimensional space (via solving the ridge problem (5) for each sample $\mathbf{X}_t$), without loss of performance compared with using the raw $Z$-values values $\mathbf{X}$. Let RAW refer to the direct prediction of targets $\mathbf{y}$ from $\mathbf{X}$, using the top 2000 most voxels most correlated with the target variable. Results of for the comparison are shown in Fig. 2(c). Only variables predicted with a a positive mean (across the different methods and across subjects) $R$-score are reported. We see that the RAW model, as expected over-fits drastically, scoring an $R^2$ of 0.3 on training data and only 0.14 on test data. Overall, for this metric CanICA performs best than all the other models in predicting the different behavioral variables on test data. However, our proposed Smooth-SODL model outperforms both SODL [16] and tCanICA, the thresholded version of CanICA.

# 7 Concluding remarks

To extract structured functionally discriminating patterns from massive brain data (i.e data-driven atlases), we have extended the online dictionary-learning framework first developed in [16], to learn structured regions representative of brain organization. To this end, we have successfully augmented [16] with a Laplacian penalty on the component maps, while conserving the low numerical complexity of the latter. Through experiments, we have shown that the resultant model –Smooth-SODL model (2)– extracts structured and denoised dictionaries that are more intepretable and better capture inter-subject variability in small medium, and large-scale regimes alike, compared to state-of-the-art models. We believe such online multivariate online methods shall become the de facto way to do dimensionality reduction and ROI extraction in the future.

**Implementation.** The authors' implementation of the proposed Smooth-SODL (2) model will soon be made available as part of the Nilearn package [2].

**Acknowledgment.** This work has been funded by EU FP7/2007-2013 under grant agreement no. 604102, Human Brain Project (HBP) and the iConnectome Digiteo. We would also like to thank the Human Connectome Projection for making their wonderful data publicly available.

## Footnotes

[1]For example, see [8, 24], implemented as part of the *Nilearn* open-source library Python library [2].

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
