[Supplementary Material]

# Supplementary material: Learning brain regions via large-scale online structured sparse dictionary-learning

**Elvis Dohmatob, Arthur Mensch, Gael Varoquaux, Bertrand Thirion**
`firstname.lastname@inria.fr`
Parietal Team, INRIA / CEA, Neurospin, Université Paris-Saclay, France

## A  Convergence of the proposed algorithm

We now show how the convergence of our proposed algorithm follows effortless from [3]. Note the the objective in (2) can be rewritten as function of the dictionary alone like so

$$\mathbb{E}_{\mathbf{x}}(\ell(\mathbf{x}, \mathbf{V})) + \gamma \sum_{j=1}^{k} \Omega_{\mathrm{Lap}}(\mathbf{V}^j) \stackrel{\mathrm{a.s}}{=} \lim_{t \to \infty} f_t(\mathbf{V}) + \gamma \sum_{j=1}^{k} \Omega_{\mathrm{Lap}}(\mathbf{V}^j),$$

where $\ell(\mathbf{x}, \mathbf{V}) := \min_{\mathbf{u} \in \mathbb{R}^k} \frac{1}{2}\|\mathbf{x} - \mathbf{V}\mathbf{u}\|_2^2 + \frac{1}{2}\alpha\|\mathbf{u}\|_2^2$, and $f_t(\mathbf{V}) := \frac{1}{t}\sum_{i=1}^{t} \ell(\mathbf{X}_i, \mathbf{V})$, with the $\mathbf{X}_i$'s sampled from the data. For each time $t \geq 0$, define

$$\hat{f}_t(\mathbf{V}) := \frac{1}{t}\sum_{i=1}^{t} \left( \frac{1}{2}\|\mathbf{X}_i - \mathbf{V}\hat{\mathbf{U}}_i\|_2^2 + \frac{1}{2}\alpha\|\hat{\mathbf{U}}_i\|_2^2 \right) + \gamma \sum_{j=1}^{k} \Omega_{\mathrm{Lap}}(\mathbf{V}^j) \qquad (1)$$

with each code $\hat{\mathbf{U}}_i$ is computed online by solving the Ridge problem (5). The following observations are immediate:

- Assumption **(A)** of [3], that the data distribution admits compact support, is automatically true for all MRI and PET data, because it is imposed by acquisition (for example, there is no data beyond the bounding-box).

- Assumption **(C)**, that the solution to the coding problem, which corresponds to (**??**)5) in our case, is unique and Lipschitz continuous w.r.t to the incoming data $\mathbf{X}_t$ is automatically satisfied for us since $\hat{\mathbf{U}}_t(\mathbf{X}_t) = (\mathbf{V}^T\mathbf{V} + \alpha\mathbf{I})^{-1}\mathbf{V}^T\mathbf{X}_t$, a linear transformation of $\mathbf{X}_t$.

As a consequence, we have the classical convergence guarantees as in in [3]:

- Almost-sure convergence of the dictionary: $\mathbf{V}(t) - \mathbf{V}(t-1) = \mathcal{O}(1/t)$ a.s. as $t \to \infty$.

- Almost-sure convergence of the risk:

  - $\hat{f}_t(\mathbf{V}(t))$ converges a.s. as $t \to \infty$.
  - $f_t(\mathbf{V}(t)) - \hat{f}(\mathbf{V}(t)) \to 0$ a.s. as $t \to \infty$.
  - $f_t(\mathbf{V}(t))$ converges a.s. as $t \to \infty$.

Finally, one notes that since constraints in problem (2) are separable in the dictionary atoms $\mathbf{V}^j$, the BCD dictionary-update algorithm Alg. 2 is guaranteed to converge to a global optimum. [1, 3].

## B Working in frequency domain, when it is possible.

To close this section, let us point out a few special instances cases of problem (7), for peculiar choices of the constraint set $Q$. First note that the objective in problem (7) can be conveniently rewritten as

$$
\begin{aligned}
F_{\gamma \mathbf{A}_t[j,j]^{-1}}(\mathbf{v}, \mathbf{V}^j + \mathbf{A}_t[j,j]^{-1}(\mathbf{V}\mathbf{A}^j - \mathbf{B}_t^j)) &= \frac{1}{2}(\mathbf{v} - \tilde{\mathbf{V}}^j)^T (\mathbf{I} - \gamma \mathbf{A}_t[j,j]^{-1}\Delta)(\mathbf{v} - \tilde{\mathbf{V}}^j) \\
&= \frac{1}{2}(\hat{\mathbf{v}} - \hat{\tilde{\mathbf{V}}}^j)^T (\mathbf{I} - \gamma \mathbf{A}_t[j,j]^{-1}\Delta)(\hat{\mathbf{v}} - \hat{\tilde{\mathbf{V}}}^j),
\end{aligned}
\tag{2}
$$

with

$$
\tilde{\mathbf{V}}^j := (\mathbf{A}_t[j,j]\mathbf{I} - \gamma\Delta)^{-1}\left(\mathbf{V}^j + \mathbf{A}_t[j,j]^{-1}(\mathbf{V}\mathbf{A}^j - \mathbf{B}_t^j)\right).
\tag{3}
$$

We note that the matrix-inversion $(\mathbf{I} - \tilde{\gamma}\Delta)^{-1}$ that appears in the formula above is a Laplacian filter, and can be efficiently applied in closed-form (i.e non-iteratively) in the Fourier / frequency domain. Indeed, under periodic boundary conditions, the discrete Laplacian $\Delta$ is Block-Circulant with Circulant Blocks (BCCB) and so is diagonalizable in the Fourier domain. Precisely,

$$
\Delta = \mathcal{F}^* \Lambda \mathcal{F}
\tag{4}
$$

where the complex orthonormal operator $\mathcal{F}$ represents the fast Discrete Fourier Transform (DFT), and $\Lambda$ is diagonal matrix made $p$ eigenvalues (including multiplicities) of the Laplace operator $\Delta$, given by

$$
\Lambda(\omega) := -\sum_{d=1}^{3}\left(2\sin\left(\frac{\omega_d \pi}{2n_d}\right)\right)^2 = -2\sum_{d=1}^{3}\left(1 - \cos\left(\frac{\omega_d \pi}{n_d}\right)\right) \leq 0,
$$

$$
\text{for } \omega = (\omega_1, \omega_2, \omega_3) \in [\![0, n_1 - 1]\!] \times [\![0, n_2 - 1]\!] \times [\![0, n_3 - 1]\!].
$$

We note that the spectral norm of Laplace operator in $D$ dimensions (here $D = 3$) is $\|\Delta\|_2 = \tilde{\gamma}_{\max}(-\Delta) = 2 \times D \times (1 + 1) = 4D$.

Now, one can then harvest the closed-form solution

$$
(\mathbf{I} - \tilde{\gamma}\Delta)^{-1}\mathbf{a} = (\mathcal{F}^{-1}(\mathbf{I} - \tilde{\gamma}\Lambda)^{-1}\mathcal{F})(\mathbf{a}) = \mathcal{F}^{-1}(\mathbf{s}),
\tag{5}
$$

where $\mathbf{s} \in \mathbb{R}^p$ is defined by $\mathbf{s}(\omega) := \dfrac{\hat{\mathbf{a}}(\omega)}{1 - \tilde{\gamma}\hat{\Delta}(\omega)}$, with $\hat{\mathbf{a}} := \mathcal{F}(\mathbf{a})$. These DFT computations have complexity $\mathcal{O}(p \log p)$.

For applying the DFTs above, one can use the FFTW[1] –or *Fastest Fourier Transform in the West*– library for computing the forward and inverse Fourier transforms needed to apply the Laplacian filter (refer to paragraph B).

**Pure $\ell_2$ constraint.** Here, the constraint set $\mathcal{C}$ is an L2 ball (with radius $= 1$, w.l.o.g) in $\mathbb{R}^2$. By the Rayleigh energy theorem (aka Parseval's identity for the DFT), one has

$$
\|\hat{\mathbf{v}}\|^2 = p\|\mathbf{v}\|_2^2, \ \forall \mathbf{v} \in \mathbb{R}^p
$$

and so problem (7) can be written as

$$
\begin{aligned}
\mathbf{V}^j \leftarrow \arg\min_{\mathbf{v} \in \mathbb{R}^p, \ \|\mathbf{v}\|_2^2 \leq 1} &\frac{1}{2}(\hat{\mathbf{v}} - \hat{\tilde{\mathbf{V}}}^j)^*(\mathbf{I} - \gamma\mathbf{A}_t[j,j]^{-1}\Lambda)(\hat{\mathbf{v}} - \hat{\tilde{\mathbf{V}}}^j) \\
&= \mathcal{F}^*\left(\arg\min_{\hat{\mathbf{v}} \in \mathbb{C}^p, \ \|\hat{\mathbf{v}}\|_2^2 \leq p}\frac{1}{2}(\hat{\mathbf{v}} - \hat{\tilde{\mathbf{V}}}^j)^*(\mathbf{I} - \gamma\mathbf{A}_t[j,j]^{-1}\Lambda)(\hat{\mathbf{v}} - \hat{\tilde{\mathbf{V}}}^j)\right) \\
&= \mathcal{F}^*\left(P_{\mathcal{E}}(\hat{\tilde{\mathbf{V}}}^j)\right)
\end{aligned}
\tag{6}
$$

where

$$
\mathcal{E} := \left\{(\mathbf{I} - \gamma\mathbf{A}_t[j,j]^{-1}\Lambda)^{\frac{1}{2}}\hat{\mathbf{v}} \text{ s.t } \hat{\mathbf{v}} \in \mathbb{C}^p, \ \|\hat{\mathbf{v}}\|_2^2 \leq p\right\},
\tag{7}
$$

a hyper-ellipsoid in standard position (i.e $\mathbf{0}$-centered and axes-aligned). Using elementary geometric arguments, one can show that the projection $P_{\mathcal{E}}(\hat{\tilde{\mathbf{V}}}^j)$ can be computed efficiently using a kind of root-finding algorithm [2], and converges exponentially fast.

**Non-negative Lasso.** In case the constraint set $\mathcal{C}$ for the dictionary atoms is a simplex $\mathcal{S}_p(\tau)$, the simplex (see section 2), then the BCD update for the $j$th atom becomes

$$\mathbf{V}^j \leftarrow \operatorname{argmin}_{\mathbf{v} \in \mathbb{R}^p, \, \mathbf{v} \geq 0, \, \mathbf{1}^T \mathbf{v} \leq 1} \frac{1}{2}(\hat{\mathbf{v}} - \hat{\tilde{\mathbf{V}}}^j)^*(\mathbf{I} - \gamma \mathbf{A}_t[j,j]^{-1}\Lambda)(\hat{\mathbf{v}} - \hat{\tilde{\mathbf{V}}}^j)$$
$$= \mathcal{F}^* \left( \operatorname{argmin}_{\hat{\mathbf{v}} \in \mathbb{C}^p, \, -\mathcal{F}^*\hat{\mathbf{v}} \leq 0, \, \hat{\mathbf{1}}^T\hat{\mathbf{v}} \leq 1} \frac{1}{2}(\hat{\mathbf{v}} - \hat{\tilde{\mathbf{V}}}^j)^*(\mathbf{I} - \gamma \mathbf{A}_t[j,j]^{-1}\Lambda)(\hat{\mathbf{v}} - \hat{\tilde{\mathbf{V}}}^j) \right), \tag{8}$$

which is a diagonal quadratic program with linear constraints, and can be effectively solved via the well-known simplex method, for example.

## C   Analytic upper bound for regularization parameter in sparse-coding

Let $\mathbf{V}$ be the current dictionary at time $t \geq 0$ (with the $t$ subscript dropped for ease of notation), and consider the sparse-coding problem

$$\mathbf{u}_t \leftarrow \operatorname*{argmin}_{\mathbf{u} \in \mathbb{R}^k} \frac{1}{2}\|\mathbf{V}\mathbf{u} - \mathbf{X}_t\|_2^2 + \alpha\|\mathbf{u}\|_1, \tag{9}$$

which is equivalent to problem (5), with the choice of penalty $\phi = \|.\|_1$ on the codes. Now, it follows from the Lasso theory that $\mathbf{u}_t$ above will be the zero vector if $\alpha \geq \alpha_{\max}(t)$, where

$$\alpha_{\max}(t) := \|\mathbf{V}^T\mathbf{X}_t\|_\infty = \max_{1 \leq j \leq p} |\langle \mathbf{V}^j, \mathbf{X}_t \rangle|$$

Now, it is clear that

$$\alpha_{\max}(t) \leq \max_{1 \leq j \leq p} \|\mathbf{V}^j\|_1 \|\mathbf{X}_t\|_\infty \leq \|\mathbf{X}_t\|_\infty \leq \sup_{t \geq 0} \|\mathbf{X}_t\|_\infty,$$

the first inequality being a consequence of the Cauchy-Schwarz and the second is due to the constraints on the dictionary atoms $\|\mathbf{V}^j\|_1 \leq 1$. Defining,

$$\|\mathbf{X}\|_{\infty,\infty} := \sup_{t \geq 0} \|\mathbf{X}_t\|_\infty := \sup_{t \geq 0} \max_{1 \leq j \leq p} |\mathbf{X}_t^j|,$$

we then obtain the rule:

$$\text{If } \alpha \geq \sup_{t \geq 0} \max_{1 \leq j \leq p} |\mathbf{X}_t^j|, \text{ then } \mathbf{u}_t = 0, \; \forall t \geq 0. \tag{10}$$

In particular, this means that a cross-validation procedure for selecting $\alpha$ only need to consider values in the range $0 \leq \alpha \leq \|\mathbf{X}\|_{\infty,\infty}$, which depends only on the input data matrix $\mathbf{X}$.

## Footnotes

[1] FFTW is generally taught to be one of the fastest implementations of the FFT, yielding up to $3\times$ speedup against competing libraries like LAPACK.