[Reviews · NeurIPS 2016]

Reviewer 1

Summary

This paper proposes a large-scale online dictionary learning framework to find interesting brain regions.

Qualitative Assessment

This paper proposes a large-scale online dictionary learning framework to find interesting brain regions. The topic itself is very interesting. However, no theoretical analysis is provided to show the effectiveness of the proposed algorithm. Moreover, the authors are not so serious for writing the paper. It seems that the authors did not read the draft carefully before the submission. For example, a comment like "146 XXX: I don’t understand the switch between V and v.” is still in the paper.

Confidence in this Review

2-Confident (read it all; understood it all reasonably well)


Reviewer 2

Summary

This paper provides an online algorithm of learning structured dictionary to characterize neuroimaging data across individuals, by extending a well-established framework in [14] with a Sobolev prior to encourage spatial contiguity of dictionary atoms. The authors applied their method on empirical data, reported the effect of varying tuning parameters and compared the performance with two alternative methods in the field.

Qualitative Assessment

Technical quality: I would rate marginally below 3 if given the option. The model definition was clear. The algorithms followed a well established framework [14], and appeared solid. However, maybe a bit undermined by its presentation, the paper did not seem to clearly demonstrate the empirical advantage of introducing the Sobolev prior in the results section. Especially, in Figure 2, it was not clear in what aspect the SSMF method was better than the two alternatives. Was the spatial boundary more smooth quantitatively? Were there certain regions, which were known to be connected, only identified by SSMF? etc. Besides (vaguely) saying results by SSMF were “more structured”, the authors could explicitly describe some criteria justified by neuroscience literature, and give quantitative measurements of the improvement according to these criteria. Added after reading the rebuttal: The results provided in the rebuttal are very informative. I changed my rating on technical quality from 2 to 3. I hope the authors include these new results in their revision, and in addition, demonstrate that the atoms learned by SSMF are more structured (with quantitative measurements) and more meaningful (with reference to neuroscience literature). Novelty/originality: There is notable original contribution in this paper, which is the extension of [14] that applied Sobolev priors to encourage spatially structured dictionary atoms. Potential impact or usefulness: Dictionary learning is a useful approach for explaining neuroimaging data across a large group of subjects, and introducing Sobolev prior or similar spatial priors should improve the results in theory. This paper provides an efficient online algorithm of doing so, and could add a good tool in the toolbox of the neuroscience/medical imaging community. The authors could also explore other applications of this method such as image segmentation and regional sensing. Presentation: The general organization of the paper is okay. However, the paper appeared not to be a well-edited manuscript; there were confusing descriptions and typos that made it harder to understand the contents. I list below the confusing parts. Some of them seem to be easy-to-fix typos, but some may require more detailed explanations. --In the matrix decomposition equation between Line 67 and 68, should the transpose symbol on the group of $V^j$ (j = 1,..., r) be on each individual $V^j$’s? --Line 73, the reference of HCP dataset was missing. --Line 83 “The Graph-Net penalty can imposes ...” --Line 146: bold sentence might be a co-author’s comment which was not fixed. --Last line of Equation (6) and Equation (8) did not agree on the sign $ V^j +A_t[j,j]^{-1}(VA_t^j -B_t^j)$ versus $A_t[j,j] V^j -(VA_t^j - B_t^j)$. Also in the last line of equation (6), I assume $V$ refers the dictionary in the last iteration, (i.e. the jth column is not the variable $v$ to optimize). -- Line 157, “a single, a single” -- Line 169, “ We did an extensive study of th how” -- In Section 3.3, the paragraph near 147 just below Algorithm 2, it would be nice if the authors explicitly gave the mathematical/algorithmic solution to Equation (6), even in the appendix. -- Figure 2, the results by different methods were presented at different horizontal and sagittal slices, which made it hard to directly compare the results. Based on the figure, I did not follow the description in Line 222 that claimed the results by SSMF model were more “structured”. -- Figure 3, it was unclear what was predicted. Was it regressing brain data on behavioral metrics, if so, was it base on $U_i$’s? The legends were not explained in the main text nor the figure caption. The figure did not seem to show the PCA baseline mentioned in the caption. I did not follow the description in Line 224-227, which claimed that R^2 greater than 0.1, (which curves should I look at?).

Confidence in this Review

2-Confident (read it all; understood it all reasonably well)


Reviewer 3

Summary

The authors extend dictionary learning models to incorporate a sobolev prior.

Qualitative Assessment

This is an interesting problem, but it is hard to judge the impact or improvement of this newly proposed algorithm. The main results figure relies on visually comparing the clarity/consistency of identified blobs on the brain, but this is not clear to an untrained eye. Also, in my experience with this type of decomposition, just as important as the algorithm (if not more) is the data. By incorporating penalties for smoothness and spatial consistency, it isn't that surprising that you get smoother blobs. Perhaps this is more useful to isolate a region with little data, but with the vast amount of data accessible with the human connectome project this doesn't seem to be the issue. I was hoping that this paper would do more to consolidate the differences of brain regions across subjects, and perhaps to use inter-subject differences/consistency as validation of the performance. Validation presented here seems very subjective and it is not clear the benefit.

Confidence in this Review

2-Confident (read it all; understood it all reasonably well)


Reviewer 4

Summary

The paper proposes under-complete dictionary learning with combined Sobolev/L1 regularization to find functional components in fMRI data.

Qualitative Assessment

The proposed approach seems interesting. The somewhat original contribution is to use dictionary learning with combined Sobolev/L1 regularization for encouraging smoothness and sparsity of the components. But the results of the new approach seem not very improved from the simpler methods, or at least, the improvement is insufficiently explained. Also, the manuscript appears quite rushed, for example there are still draft comments (line 146) and it is a quite tedious, sometimes even confusing read.

Confidence in this Review

2-Confident (read it all; understood it all reasonably well)


Reviewer 5

Summary

This paper focuses on the dictionary learning in the application of brain regions learning. Concretely, the authors propose a large-scale online dictionary learning framework with an additional Sobolev regularization for the structure.

Qualitative Assessment

This paper focuses on the dictionary learning in the application of brain regions learning. Concretely, the authors propose a large-scale online dictionary learning framework with an additional Sobolev regularization for the structure. My major comments are as follows. 1) As far as I am concerned, this paper merely extends the online dictionary learning (J Mairal, JMLR2010) with an additional Sobolev regularization term. Thus I think the novelty is fairly weak and fails to cohere with the NIPS standard. 2) In order to impose structure constraint on the learned dictionary, this paper utilizes Sobolev regularization. However, the detailed definition of “structure” is not very clear. If the structure means the sparseness, why the sparseness of dictionary itself would bring significant improvement in your problems? All in all, the motivation of the usage of Sobolev is not clear. 3) I wonder whether there is a mistake in Eq.(1); it should be [V^1,V^2,…,V^r]^T. 4) More statement and analysis of the construction of Eq.(2) is needed. For example, why is the l2 norm imposed on the codes instead of l1 norm in (J Mairal, JMLR2010)? 5) What is the line 146? 6) The language of this paper needs to be refined. I have found many typos in it.

Confidence in this Review

2-Confident (read it all; understood it all reasonably well)


Reviewer 6

Summary

A new regularization for matrix factorization is introduced to impose sparsity and smoothness. Although the existence of the solution is not shown, the algorithm of obtaining a solution is presented. It is demonstrated experimentally that that the result is superior to the stat-of-the-art approach.

Qualitative Assessment

Introduction of a new reqularizer in the optimization problem is a strong side of the paper. The results of the experiments show the potential of the approach. However as the author(s) states the problem is non-convex. The existence of the solution is not shown. The algorithm is solely backed up by experiments. Another issue is that the written needs improvement, say line 146. RE: after rebuttal I am satisfied with the reply on the conference of the algorithm. I am willing to change my rate on Clarity and presentation from 2 to 3.

Confidence in this Review

3-Expert (read the paper in detail, know the area, quite certain of my opinion)